



# A Novel Framework for Automatic Scanning Radar Pointing Calibration Using the Sun

Paul Ockenfuß[1], Gregor Köcher[1], Matthias Bauer-Pfundstein[2], and Stefan Kneifel[1]

[1]Meteorologisches Institut, Ludwig-Maximilians-Universität München, Germany
[2]METEK Meteorologische Messtechnik GmbH

**Correspondence:** Paul Ockenfuß (paul.ockenfuss@physik.uni-muenchen.de)

**Abstract.**

A high accuracy of antenna beam pointing is essential for weather and cloud radars in order to precisely locate cloud and precipitation. It is also a critical requirement for estimating the horizontal wind field or to retrieve particle's vertical motions.

We present a general framework for radar pointing calibration using the sun as a reference target. The workflow is structured into three steps: (i) measurement and analysis of individual Sun scans, (ii) estimation of scanner inaccuracies from a series of scans, and (iii) correction of these inaccuracies. Our approach is radar-agnostic and applicable to any instrument equipped with a two-axis pan-tilt scanner and a parabolic antenna. General recommendations for Sun scan implementation are given, and the full calibration process is demonstrated using a Mira-35 cloud radar. The method allows retrieval of a comprehensive set of parameters, including beamwidth in two orthogonal directions, pedestal tilt, axis misalignments, encoder offsets, gear backlash, and the receiver-scanner time offset. With this approach, absolute pointing accuracy better than $0.1°$ can be achieved, and relative changes as small as $0.01°$ can be detected. To facilitate the automatic application, we provide the open-source Python library `SunscanPy` for radar pointing calibration. This toolset is especially valuable for stationary radars and radar networks, where it enables automatic monitoring of long-term pointing stability. Finally, we introduce a novel automatic pointing correction scheme based on inverse kinematics. Once the scanner inaccuracies are estimated, the required motor positions can be computed to compensate for the inaccuracies, without mechanical adjustments. Such functionality is particularly advantageous for mobile radars, research campaigns, or remote deployments, where frequent mechanical leveling is necessary but often difficult to perform.

## 1 Introduction

Accurate antenna pointing is crucial for a wide range of radar applications. Applications that rely on precise georeferencing, such as scanning weather radars, are particularly sensitive, since even small pointing errors lead to spatial displacements of observed features. For example, a $1°$ change in pointing direction leads to $3.5\,\mathrm{km}$ displacement in a distance of $200\,\mathrm{km}$. For vertically pointing radars, pointing accuracy is equally critical for the correct measurement of vertical velocities. Since vertical air motion and particle sedimentation speeds are typically an order of magnitude smaller than horizontal wind, even small pointing errors can lead to large biases. A simple calculation illustrates this: assuming a horizontal wind speed of $30\,\mathrm{m\,s^{-1}}$, as



is common in the higher atmosphere, a mispointing of $0.2°$ produces an apparent vertical velocity of $\pm 10.5\,\mathrm{cm\,s^{-1}}$, depending on the wind direction. This is a non-negligible error, given the typical sedimentation velocity of ice around $30\,\mathrm{cm\,s^{-1}}$ to $50\,\mathrm{cm\,s^{-1}}$. In principle, modern Doppler radars are capable of measuring velocities with a precision on the order of $1\,\mathrm{cm\,s^{-1}}$. Furthermore, precise pointing is also indispensable in advanced applications such as multi-frequency measurements, where two radars must probe the same volume simultaneously, which demands near-perfect beam alignment (Kneifel et al., 2016;
Tridon and Battaglia, 2015).

  To ensure accurate pointing, a variety of calibration methods have been developed. One straightforward approach is the use of hard targets at a known position or ground-clutter (e.g. Altube et al., 2016). While effective in principle, hard targets are not always available, probe only one specific scanner configuration, and ground-clutter based methods may be compromised by precipitation or anomalous atmospheric propagation conditions (Altube et al., 2016). A widely adopted alternative is the Sun,
which acts as a natural microwave source. The advantage of Sun-based calibration is that it requires minimal additional effort: operational radar measurements occasionally intersect the solar disk by chance, and these data can be used even without specific campaigns. For scanning research weather radars, dedicated solar scans are often performed as part of calibration routines. The benefits of solar calibration are now widely acknowledged, and some radar manufacturers have integrated corresponding procedures into their operational software.

A number of studies have investigated the use of the Sun for radar calibration, with different emphases on pointing accuracy and antenna characterization. Early work by Baars (1973) discussed cosmic sources such as the Sun for large antenna calibration. In addition to antenna pointing, other antenna parameters, such as aperture or antenna beam pattern, were derived from measurements on radio sources. Later, Mano and Altshuler (1981) discussed the Sun as a potential calibration target with a well-known position, focusing primarily on elevation biases. In a review of radar calibration methods, also Manz et al. (2000)
highlight the Sun as an excellent source to calibrate position accuracy. Arnott et al. (2003) demonstrated that solar observations could also be used to estimate the azimuth orientation of a mobile radar system mounted on a truck. Similarly, Darlington et al. (2003) explored the potential of solar signals for calibrating operational weather radars in the United Kingdom. Their study emphasized the advantage of performing the calibration without disturbing the ongoing operational scan strategy. Due to the limited maximum elevation angle of their radar ($4°$), their analysis was restricted to azimuth biases. Further develop-
ments were made by Huuskonen and Holleman (2007) and later by Holleman et al. (2010), who refined the methodology for operational weather radars. In particular, they introduced corrections for atmospheric refraction, which is especially relevant for low-elevation measurements where the apparent solar position is significantly displaced. Muth et al. (2012) proposed a technique based on the analogy between radar and theodolite systems, using solar observations to quantify and correct pointing errors in both azimuth and elevation. Their evaluation of a single Sun scan is based on Huuskonen and Holleman (2007). More
recently, Altube et al. (2015) demonstrated how the Sun can be used for online monitoring of the antenna alignment of an operationally scanning weather radar. In a subsequent study, Altube et al. (2016) then compared ground-clutter-based and sun-based antenna pointing calibration methods and found the methods based on the Sun to be generally superior. Recently, Frech et al. (2019) assessed the antenna pointing accuracy in both, azimuth and elevation, for a polarimetric research radar based on dedicated solar box scans. In addition to pointing, the Sun has also been used to characterize the antenna beam pattern.



Reimann and Hagen (2016) employed solar scans to measure the antenna beam shape and demonstrated good agreement with the expected response to a point source. While our work focuses on antenna pointing calibration, the beam pattern retrieval comes as a natural by-product of the same procedure.

Although these studies have shown in the past that the Sun provides an excellent target for antenna pointing calibration, two important deficiencies remain. First, there is no general, radar-agnostic framework for processing Sun scan data. While

manufacturers provide proprietary tools, these are often tailored to specific radar systems and are not easily transferable. Yet, the underlying geometry is universal: virtually all weather and cloud radars operate on a two-axis pan-tilt mechanism, and once the solar data are recorded, the processing steps are conceptually identical across platforms. Second, existing calibration workflows typically assume that mechanical correction of the scanner is required once biases are identified. We argue that an alternative is possible: active, software-based correction of mispointing. By interpreting the radar scanner as a simple kinematic chain

with two rotational joints, the problem is directly analogous to inverse kinematics in robotics, where the controller software compensates for misalignments instead of mechanically shifting the system.

The objective of this publication is therefore twofold. First, we present a novel, generic framework for scanning radar pointing calibration, subdividing the process into three well defined, consecutive steps. By combining multiple Sun scans, our method is able to derive the full set of scanner offsets and inaccuracies in both axes, including velocity-dependent dy-

namic biases. To make this methodology applicable by the broader community, we provide `SunscanPy`, a general and open-source Python package for radar antenna pointing calibration using solar observations. The software is designed to be radar-independent and provides a standardized way to derive scanner inaccuracies. Second, we demonstrate a method for software-based pointing correction, in which identified biases are incorporated into the radar controller, thereby avoiding the need for mechanical realignment.

The remainder of this publication is organized as follows. section 2 describes the theoretical background of antenna pointing calibration using the sun and introduces a mathematical nomenclature. section 3 demonstrates the process using measurements of a cloud radar. Both sections contain three dedicated subsections, reflecting the three steps of the calibration method. The achievable pointing accuracy of the method is discussed in subsection 3.4. section 4 introduces the `SunscanPy` Python implementation. section 5 summarizes the results and provides perspectives on possible areas of application.

## 2  Methods

The whole process of scanner calibration with the Sun can be divided into three distinct steps, as illustrated in Figure 1. In step 1, we use the Sun as a source of microwave emission in the sky with a precisely defined trajectory. Probing this source with the radar antenna, we can record the position of the scanner axis encoders when looking at a defined position relative to the sun, e.g. the center of the solar disk. Repeating step 1 over the course of multiple days and times, we obtain a collection of scanner

positions with the corresponding beam locations in the sky. In step 2, we use this dataset to estimate unknown parameters that define the scanner system. In step 3, we apply the knowledge of these parameters when pointing to a target. We can either correct the scanner mechanically like illustrated in step 3a, or do a software based correction like depicted by step 3b.



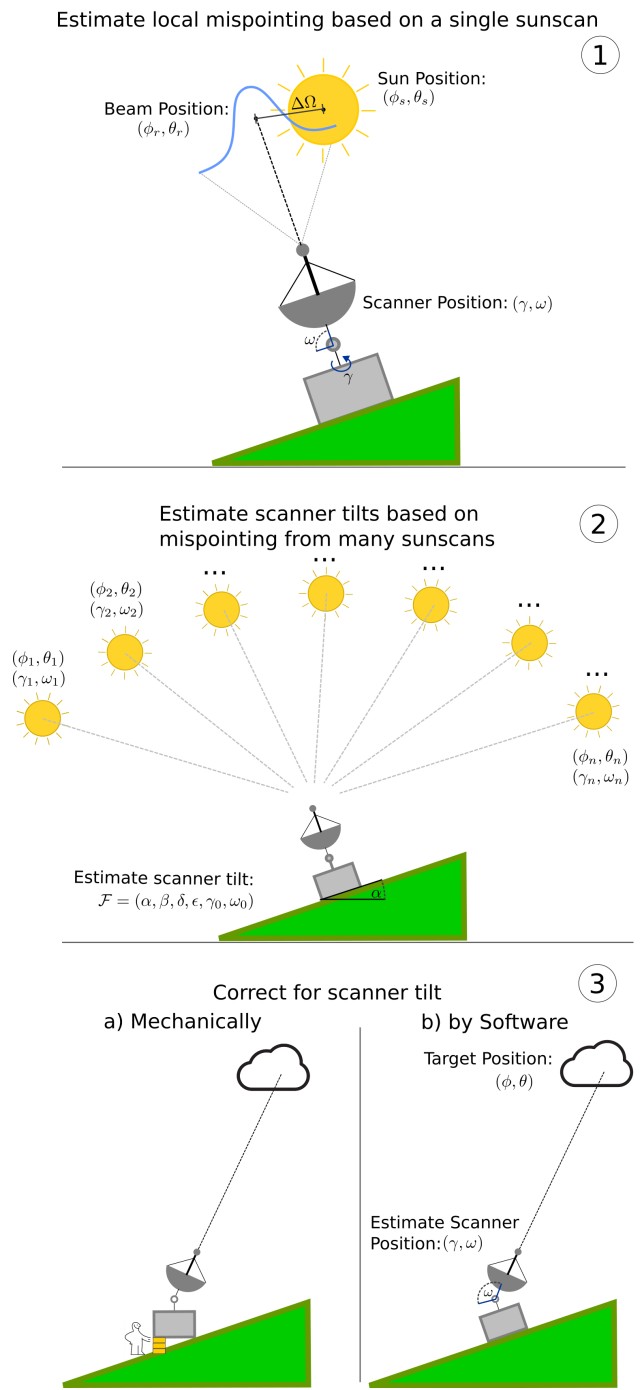

**Figure 1.** Schematic illustration of the full pointing calibration procedure. Step 1: Estimation of local mispointing from a single Sun scan. Step 2: Estimation of scanner inaccuracies based on multiple Sun scans. Step 3: Correction of scanner inaccuracies.





## 2.1 Scanner Inaccuracies

Pointing directions in the sky can be described by two spherical coordinates, where $\phi \in [0°, 360°]$ denotes the azimuth angle and $\theta \in [0°, 90°]$ is the elevation angle. We follow the common convention in radar science that 0º elevation corresponds to pointing horizontally. Scanning radars usually have a two-axis scanner mechanism, where one axis corresponds to the movement in azimuth direction and another axis on top of the first moves in elevation direction. We denote the position of those axes by $\gamma \in [0°, 360°]$ and $\omega \in [0°, 180°]$, respectively. Usually, it is assumed that a movement in $\gamma$ directly corresponds to a movement in $\phi$ and similar for $\omega$ and $\theta$. However, this is only the case for a perfect scanning mechanism. In reality, axis positions are not perfectly equal to pointing directions, due to multiple system imperfections: Axis encoders can have offsets, the axes are not perfectly orthogonal, or the whole system is tilted with respect to the celestial hemisphere. Therefore, a strict separation between axis positions $(\gamma, \omega)$ and celestial pointing $(\phi, \theta)$ is necessary. We describe the mapping between axis positions and celestial pointing of the radar by a forward model $M_{\mathcal{P}}$:

$$M_{\mathcal{P}} : (\gamma, \omega) \mapsto (\phi, \theta)$$

Finding $M_{\mathcal{P}}$ is the main goal of the methods described in here. $M_{\mathcal{P}}$ depends on multiple parameters $\mathcal{P}$, which describe the imperfections of the scanner alignment and mechanics. There are seven static parameters:

$$\mathcal{P} = (\alpha, \beta, \delta, \epsilon, \gamma_0, \omega_0, \chi) \tag{1}$$

The individual static scanner parameters are illustrated in Figure 2 and defined as follows:

- $\alpha$: Pedestal tilt towards West or East

- $\delta$: Pedestal tilt towards North or South

- $\epsilon$: Ideally the radar beam should be perpendicular to the axis of the elevation rotation of the scanner. $\epsilon$ describes the offset from this goal.

- $\beta$: Gimbal tilt. Ideally the axis of the elevation and axis of the azimuth rotations of the scanner should be perpendicular. $\beta$ describes the offset from this goal. The gimbal tilt $\beta$ and the antenna tilt $\epsilon$ have the same effect if the scanner is pointed vertically or close to vertically. If the beam is pointing to horizon or close to horizon then the antenna tilt $\epsilon$ causes an azimuth offset but the effect of the gimbal tilt $\beta$ vanishes.

- $\gamma_0$: Offset between actual azimuth axis position and the value returned by the scanner motor encoder. This value is sometimes known as the "northangle".

- $\omega_0$: Offset between actual elevation axis position and the value returned by the scanner motor encoder.

- $\chi$: If not pointing vertical, the weight of the antenna can cause elastic deformations, which cause the scanner structure to bend. This causes an elevation offset, which depends on the current elevation position. We model this effect as an elevation position offset of the form $\omega_{\text{flex}} = \chi \cos(\omega)$. $\chi$ is usually negative.





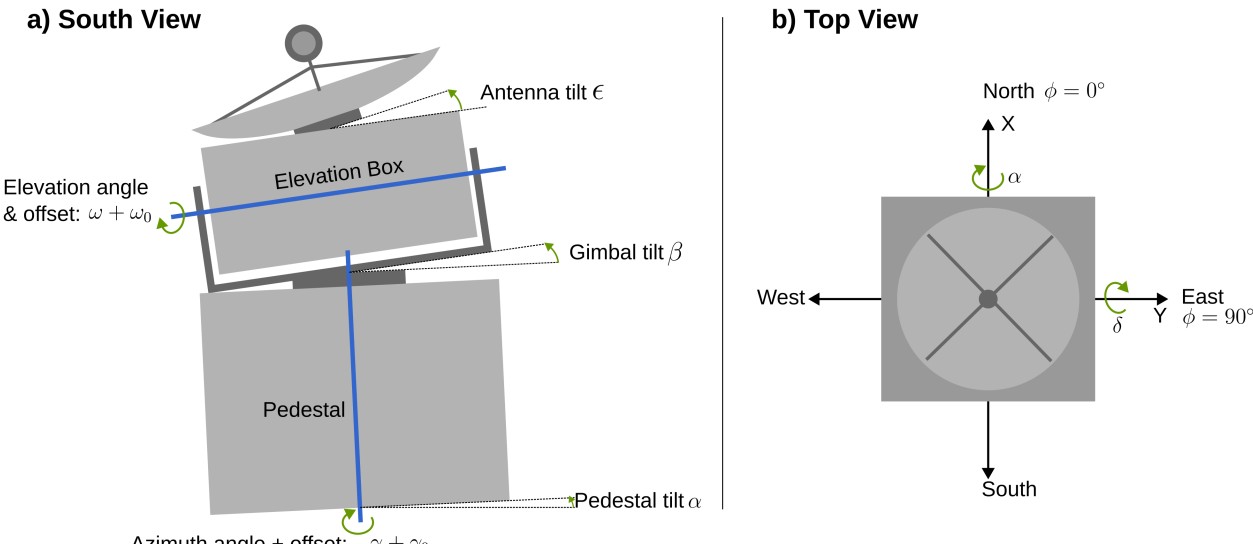

**Figure 2.** Inaccuracies of the scanner leading to beam pointing offsets, with the axes positioned at $\gamma = 0°$ and $\omega = 90°$. **a)**: View from the South. **b)**: View from the top. The sign convention of each parameter is indicated by green arrows, which point into the positive direction of movement. Positive $\alpha$ tilts the radar West, positive $\delta$ tilts the radar North. Positive $\gamma$ tilts the dish clockwise, if viewed from the top. For $\gamma = 0°$ and $\omega = 0°$, the dish points to the North. For $\gamma = 0°$, positive $\omega$ tilts the elevation box clockwise, if viewed from the West.

We denote a perfectly accurate scanner as $M_{\mathcal{I}}$, with $\mathcal{I} = (0,0,0,0,0,0,0)$. For such a perfect scanner, axis positions and celestial pointing are identical (at least, as long as $\omega <= 90°$): $\gamma = \phi$ and $\omega = \theta$.

In addition to the static scanner parameters, there are also dynamic parameters. Dynamic means that their effect on the beam pointing depends on the velocity of the axes ($\dot{\gamma}$, $\dot{\omega}$). We consider the following dynamic parameters:

- $b_\gamma$: backlash in the azimuth axis. For example, this can be caused by play in the gears of the azimuth axis. As a consequence, this creates a hysteresis in the system, where the beam is always lagging behind the position indicated by the scanner motors. The backlash only depends on the direction of movement. The effect of the backlash on the actual position $\tilde{\gamma}$ of the azimuth axis can be modeled as $\tilde{\gamma} = \gamma + b_\gamma \text{sign}(\dot{\gamma})$

- $t_0$: time offset between the time of recording of the motor positions and the time of recording of the signal. It causes a lag between motor positions and beam, but in contrast to the lag caused by the backlash, the lag due to the time offset depends on the speed of motion. In the absence of acceleration of the motors, the actual positions can be calculated as $\tilde{\gamma} = \gamma + t_0\dot{\gamma}$ and $\tilde{\omega} = \omega + t_0\dot{\omega}$.

We assume that the effect of the dynamic parameters is the same, regardless of the scanner pointing position in the sky. Therefore, they can simply be modeled as offsets to the scanner coordinates:

$$M_{\mathcal{P}}(\gamma + b_\gamma \text{sign}(\dot{\gamma}) + t_0\dot{\gamma},\ \omega + t_0\dot{\omega}) \tag{2}$$




| Step | 1. Local Mispointing | 2. Scanner Inaccuracies | 3. Inverse Kinematics |
|---|---|---|---|
| Optimized parameters | - Local mispointings $\Delta\gamma$, $\Delta\omega$<br>- Beamwidths $f_x$, $f_y$<br>- Backlash in azimuth $b_\gamma$<br>- Time offset $t_0$ | - Pedestal tilts $\alpha$, $\delta$<br>- Axis offsets $\gamma_0$, $\omega_0$<br>- Gimbal tilt $\beta$<br>- Antenna tilt $\epsilon$<br>- Elastic deformation $\chi$ | Scanner positions $\gamma$, $\omega$ |
| Target | Measured signal strengths $S_i$ | Reference beam positions $\phi_r, \theta_r$ | Celestial position $\phi, \theta$ |
| Forward Model | Simulated signal strengths $Q_i$ | Scanner model $M_\mathcal{F}$ (kinematic chain) | Scanner model $M_\mathcal{F}$ |
| Input for forward model | Time $t_i$, scanner positions $\gamma_i$, $\omega_i$ and scanner velocities $\dot{\gamma}_i$, $\dot{\omega}_i$ | Reference axis positions $(\gamma_r, \omega_r)$ | Scanner positions $\gamma, \omega$ |
| Objective Function | RMSD between $S$ and $Q$ (in dB units) | RMSD of mispointing $\Delta\Omega$ between forward simulated and target beam position | RMSD of mispointing $\Delta\Omega$ between estimated and target position |

**Table 1.** Overview of the optimized parameters in each step. The steps are illustrated in Figure 1.

## 2.2 Step 1: Estimate Local Mispointing

If the Sun were a perfect point source and the radar emitted an infinitely narrow beam, step 1 would not be necessary to get a pair of referenced scanner coordinates. In that idealized case, we could simply scan across the sky until the beam intersects the point-like Sun, record the corresponding scanner coordinates, and directly assign them to the absolute solar position.

In reality, the Sun appears as an extended, moving disk and the radar beam exhibits a finite width. Consequently, the measured signal represents a broadened response, and we must determine the apparent center of the Sun within the data. Similar to Huuskonen and Holleman (2007), we will do so by fitting a simulated response to all data points. In addition, we will also obtain information about the radar beam shape, scanner backlash and signal time offset. To achieve this, we perform multiple measurements in the region where the Sun is expected to be located. This procedure yields a set of $N$ samples, each containing the measurement time, scanner coordinates and velocities, and recorded signal:

$$\text{Sample i} : (t_i, \gamma_i, \omega_i, \dot{\gamma}_i, \dot{\omega}_i, S_i). \tag{3}$$

According to the measurement time $t_i$, we compute the apparent solar position $(\phi_{s,i}, \theta_{s,i})$, accounting for the radar position and altitude.

If the scanner operated without inaccuracies, the corresponding celestial beam position at time $t_i$ would be given by the ideal mapping

$$M_\mathcal{I}(\gamma_i, \omega_i) = (\phi_{b,i}, \theta_{b,i}). \tag{4}$$





However, with a single Sun scan it is not feasible to determine the full set of inaccuracy parameters $\mathcal{P}$ of a general scanner model $M_{\mathcal{P}}$, since the observations cover only a small portion of the sky. We therefore restrict the model to two parameters

$(\Delta_\gamma, \Delta_\omega)$, which capture the local mispointing of the system in the scanned region. In this approximation, the effective beam direction is

$$M_{\mathcal{I}}(\gamma_i + \Delta\gamma, \omega_i + \Delta\omega) = (\phi_{b,i}, \theta_{b,i}). \tag{5}$$

To estimate $(\Delta\gamma, \Delta\omega)$, we simulate the expected signal strength $Q(\phi_b, \theta_b, \phi_s, \theta_s)$ as a function of beam and solar position, as well as several other parameters listed in Table 1. The optimal parameters are then obtained by minimizing the root-mean-

squared deviation (RMSD) between measured and simulated signals:

$$\min_{\text{(Q Parameters)}} \left( \sqrt{\frac{1}{N} \sum_{i=1}^{N} (S_i - Q_i)^2} \right). \tag{6}$$

With $\Delta\gamma$ and $\Delta\omega$ available, we can finally calculate a reference pair of scanner coordinates $\gamma_r, \omega_r$ with a reference position in the sky $\phi_r, \theta_r$.

$$M_{\mathcal{I}}(\gamma_r + \Delta\gamma, \omega_r + \Delta\omega) = (\phi_r, \theta_r). \tag{7}$$

$\gamma_r$ and $\omega_r$ can be chosen arbitrarily from all samples $(\gamma_i, \omega_i)$.

### 2.2.1 Sun Scan Signal Simulation

Given a beam and Sun position, the expected signal in the receiver is determined by the convolution of the antenna's receiving pattern with the Sun's emission pattern. By the reciprocity theorem of electromagnetics, the receiving and transmitting patterns of the antenna are identical. In the following, we therefore refer to both as the radar beamshape.

Huuskonen and Holleman (2007) assume that the convolved signal distribution follows a Gaussian function. This assumption is valid, if the beam opening angle is significantly broader than the sun opening angle, but leads to inaccuracies for narrower beams. Therefore, we numerically simulate the expected signal based on the convolution of a discretized version of the sky and antenna patterns.

We assume that the Sun and beam positions differ only by a small angular offset $\pm\nu$. [1] This allows to approximate the

celestial hemisphere as locally flat and to transform both positions into a Cartesian reference frame centered on the beam direction (illustrated in Figure 3).

The beam-centered axes are defined as follows: the local $b_z$ axis points radially outward along the beam, the local $b_x$ axis lies orthogonal to $b_z$ and $e_z$ in the cross-elevation direction, and the local $b_y$ axis completes the left-handed system,

---

[1]Note that in reality, this assumption does not limit the applicability of the method. Even though our Mira35 radar has a northangle of 202°, by choosing an appropriate initial guess, the optimizer is never confronted with deviations between Sun and expected beam position of more than about 5°. `SunscanPy` will find such an initial guess automatically before starting the optimization.



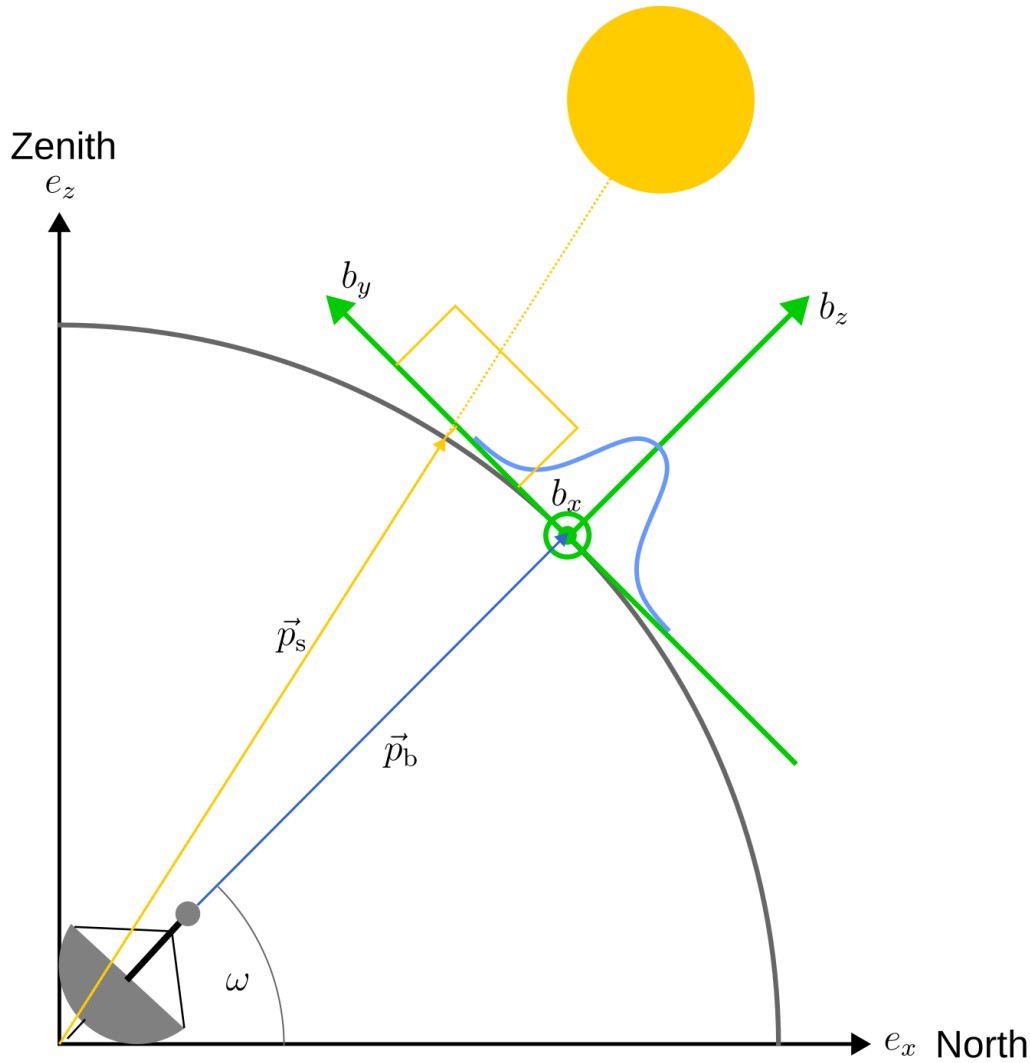

**Figure 3.** 2D illustration of the definition of the beam centered coordinate frame (green). The radar beam pointing and antenna pattern are illustrated in blue, the sun pointing vector and emission pattern are illustrated in yellow.

$$b_z = \begin{pmatrix} \cos\phi_b \cos\theta_b \\ \sin\phi_b \cos\theta_b \\ \sin\theta_b \end{pmatrix}, \qquad b_x = \frac{e_z \times b_z}{|e_z \times b_z|}, \qquad b_y = b_z \times b_x. \tag{8}$$





The beamshape of a parabolic antenna can be expressed by a two-dimensional Airy pattern (Kraus and Marhefka, 2008). In the beam-centered system, this is described by Equation 9, which contains the first order Bessel function $J_1$.

$$G(r) = G_0 \left( \frac{2 J_1(r)}{r} \right)^2, \qquad r = \sqrt{\left( \frac{x}{x_0} \right)^2 + \left( \frac{y}{y_0} \right)^2} \tag{9}$$

The coefficients in Equation 9 are defined as

$$x_0 = \frac{f_x}{2 r_{0.5}}, \qquad y_0 = \frac{f_y}{2 r_{0.5}}, \qquad r_{0.5} = 1.6163399 \tag{10}$$

$f_x$ and $f_y$ are the beam full widths at half maximum (FWHM) in the $x$ and $y$ directions, respectively, $r_{0.5}$ it the value where the main lobe has half the maximum value: $G(r_{0.5}) = \frac{G_0}{2}$. $G_0$ is chosen such that the integral over $G$ is one.

The solar emission pattern is modeled as a simple bipartite function,

$$H(x,y) = \begin{cases} H_0 & \text{if } (x,y) \text{ lies outside the solar disk,} \\ H_1 & \text{if } (x,y) \text{ lies inside the solar disk,} \end{cases} \tag{11}$$

where $H_0$ is the background sky brightness and $H_1$ the solar disk brightness. The Sun is represented as a circle of angular radius

$$r_s = \arcsin\left( \frac{r_{\text{planetary}}}{d_{\text{S-E}}} \right), \tag{12}$$

with planetary solar radius $r_{\text{planetary}} = 695{,}660$ km (Haberreiter et al., 2008). The Sun-Earth distance $d_{\text{S-E}}$ varies by about 3 % throughout the year due to Earth's orbital eccentricity and is evaluated for the scan time using the `Skyfield` library (Rhodes, 2019). At 35 GHz, the electromagnetic thermal emission of the sky is at least 13 dB lower than the emission of the Sun at 5800 K. Since for the Mira35 system, the Sun emission is around 4 dB higher than the noise floor of the receiver, this sky emission can not be detected and is set to $H_0 = 0$. Figure 4 illustrates $G(x,y)$ and $H(x,y)$ in the beam-centered coordinate system.

The simulated signal $Q$ is then the convolution of the solar emission function $H$ and the beamshape $G$ plus the receiver noise $H_n$:

$$Q = H * G + H_n = H_1 \int_{\text{sundisk}} G(x,y)\,dx\,dy + H_n \tag{13}$$

To evaluate $Q$, both $H_n$ and $H_1$ must be determined. The receiver noise $H_n$ can be measured directly by pointing the antenna sufficiently far from the Sun, i.e. by at least one expected beamwidth plus the pointing uncertainty. The solar brightness $H_1$, however, cannot directly be measured. Instead, it is inferred from the maximum measured signal $S_{\text{max}}$. Assuming that $S_{\text{max}}$ originates from a beam direction close to the center of the solar disk, we can solve Equation 13 for $H_1$:

$$H_1 = \frac{S_{\text{max}} - H_n}{\int_{\text{centered sundisk}} G\,dx\,dy} \tag{14}$$

This approach implies that the simulated signal at the disk center is by construction equal to $S_{\text{max}}$. Thus, the beam shape influences only the relative decay of the signal from the Sun's center towards the edge. A narrow beam produces a sharp transition between Sun and sky, while a wide beam yields a smoother decay.

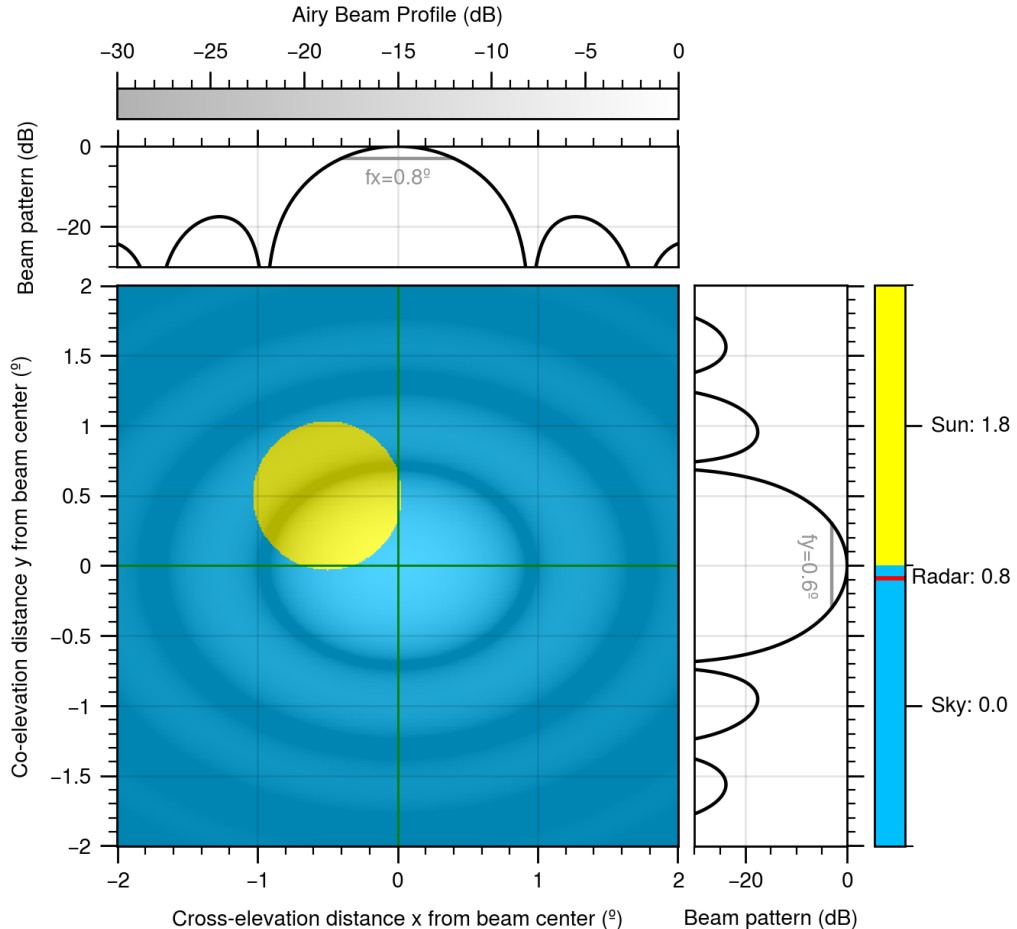

**Figure 4.** Illustration of the sun and sky emission pattern in beam centered coordinates, as "seen" by the radar. The background consists of just two emission values: The sun (yellow) and the sky around (blue). This background is overlayed with an airy beampattern with $f_x = 0.8$, $f_y = 0.6$. The beampattern is illustrated by brighter and darker regions in the image. Darker means the radar is insensitive in these regions. The top grey-white colorbar shows the sensitivity in dB. Panels b) and c) show horizontal and vertical cross sections of the beampattern. The expected signal detected by the radar is the summed product of background and beam pattern plus receiver noise. It yields a value in between the brightness of the puse sun and pure sky, as shown by the red line in the right colorbar.

### 2.2.2 Proposed Scanning Pattern

Before conducting a Sun scan, we recommend adjusting the radar scanner to a moderate pointing precision, e.g. $\nu = 2°$. This avoids the need to "search" the Sun across the entire celestial hemisphere. The adjustment can usually be performed visually or with a simple spirit level and compass. Once this preliminary alignment is completed, multiple samples are recorded within



a region of angular size $2\nu$ around the expected Sun position:

$$\gamma \in [\phi_s - \nu, \ \phi_s + \nu], \qquad \omega \in [\theta_s - \nu, \ \theta_s + \nu]. \tag{15}$$

The specific scanning pattern used within this region is not critical, since all sample tuples $(t_i, \gamma_i, \omega_i, \dot{\gamma}_i, \dot{\omega}_i, S_i)$ are treated independently. In practice, rectangular or rhombus-shaped "zigzag" scans are often employed. However, any pattern is acceptable provided that the following criteria are satisfied:

1. **Coverage of the solar disk.** The data must contain some measurements with the beam probing the solar disk's central region. If this is not the case, the search region $\nu$ must be enlarged.

2. **Measurement of background sky noise.** At least one sample must be taken sufficiently far from the Sun to record the receiver noise $H_n$. As a rule of thumb, this can be achieved at an azimuth offset of $\nu + f_x$, i.e. the uncertainty in radar pointing plus the (maximum) expected beamwidth.

3. **Use of multiple azimuth velocities.** Ideally, the scan should include at least two different azimuth velocities. Since mechanical backlash depends only on the velocity sign, this enables the setup of two independent equations, which 215 allow simultaneous estimation of both the backlash $b_\gamma$ and the timing offset $t_0$:

$$\begin{aligned}
b_\gamma \, \text{sign}(\dot{\gamma}_1) + t_0 \dot{\gamma}_1 &= b_\gamma + t_0 \dot{\gamma}_1, \\
b_\gamma \, \text{sign}(\dot{\gamma}_2) + t_0 \dot{\gamma}_2 &= b_\gamma + t_0 \dot{\gamma}_2.
\end{aligned} \tag{16}$$

Apart from these requirements, the absolute axis velocities, sampling rate, and ordering of samples may be chosen freely according to the capabilities of the scanner and the signal processing system. Appendix A presents our specific scan pattern in the form of pseudocode.

**2.2.3 Reverse Scanning**

Many scanner systems permit elevation angles greater than $90°$. In such cases, most points on the celestial hemisphere can be reached by the scanner in two different configurations: a forward configuration with $\omega \leq 90°$ and a reverse configuration with $\omega > 90°$.

Assuming an ideal scanner $M_{\mathcal{I}}$, forward and reverse positions are related by a rotation of $180°$ in azimuth combined with 225 the complementary elevation:

$$\gamma' = \gamma + 180°, \qquad \omega' = 180° - \omega. \tag{17}$$

Performing sun scans in both forward and reverse configurations is advantageous because it allows to resolve ambiguities in the scanner parameters. For instance, consider the case where a positive elevation mispointing $\Delta\omega$ is observed, meaning the radar is pointing above the Sun. This deviation could originate either from an elevation axis offset or from a tilt of the scanner 230 pedestal. In the reverse configuration, however, the two effects manifest differently: a pedestal tilt still produces a positive $\Delta\omega$,



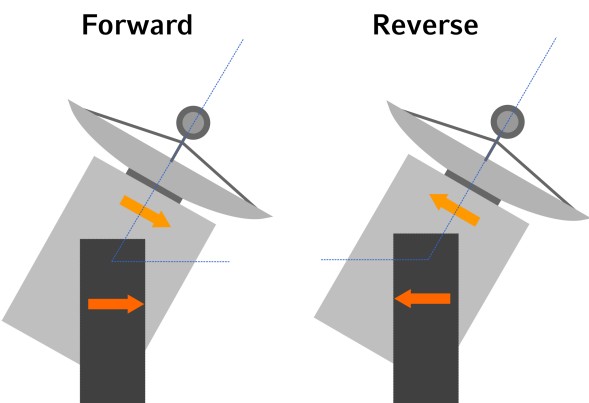

**Figure 5.** Illustration of the forward and reverse configuration of a two axis scanner. Note that in the reverse configuration, the scanner has turned by $180°$ in azimuth.

whereas an elevation offset causes a sign flip, leading the radar to point below the Sun. The reverse scan must satisfy the same requirements as the forward scan regarding coverage of the solar disk, measurement of sky noise, and inclusion of multiple scanning velocities.

### 2.3 Step 2: Estimate Scanner Inaccuracies

A single Sun scan provides information only about the local mispointing in one specific region of the sky. However, in other regions the mispointing may differ, depending on the specific inaccuracies of the scanner introduced in Sec. 1. To estimate these inaccuracies systematically, we implemented the forward model $M_{\mathcal{P}}(\gamma, \omega)$ as a kinematic chain using `ikpy`, a Python library for robotics and inverse kinematics (Manceron, 2024). The kinematic chain consists of four links: Origin, Azimuth joint, Elevation joint, and Antenna. Each link defines a rotated coordinate frame relative to the previous one, with the rotations depending on the parameters $\alpha$, $\nu$, $\beta$, $\epsilon$, $\tilde{\gamma}$, and $\tilde{\omega}$. The effects of backlash $b_{\gamma}$, time offset $t_0$, angle offsets $\gamma_0$ and $\omega_0$, as well as structural flexibility $\chi$, are combined into the effective angles $\tilde{\gamma}$ and $\tilde{\omega}$.

A single Sun scan is not sufficient to unambiguously determine all parameters in $\mathcal{P}$. For example, when scanning the Sun in the South, no information can be inferred about a potential East-West pedestal tilt $\alpha$. To resolve such ambiguities, Sun scans at multiple positions across the sky are required, ideally covering the course of a full day in the summer months. Each Sun scan
yields a pair of referenced coordinates $(\gamma_r, \omega_r)$ in the scanner system and the corresponding celestial reference coordinates $(\phi_r, \theta_r)$.

For a given set of estimated parameters $\mathcal{F}$, we define an objective function based on the mispointing angle between the predicted beam position and the celestial reference position:

$$\Delta\Omega\big(M_{\mathcal{F}}(\gamma_r, \omega_r), (\phi_r, \theta_r)\big). \tag{18}$$



The objective is defined as the RMSD of the mispointing across all referenced pairs:

$$O = \sqrt{\frac{1}{N} \sum_{i=1}^{N} \Delta\Omega_i^2}. \tag{19}$$

Minimizing this objective yields the set of parameters $\mathcal{F}$ that best explains the observed radar pointing across all Sun scans. To carry out the minimization, we employ a two-step strategy: first, a three-point brute-force search provides reasonable starting values; second, a Nelder-Mead simplex optimization minimizes the objective.

Global optimization in a high-dimensional parameter space is challenging, and simultaneous fitting of all parameters can cause the optimizer to become trapped in local minima. We found that sequential fitting, leveraging the specific mispointing signatures of each parameter, avoids this issue. In our approach, previously fitted parameters are held fixed as best guesses in subsequent steps:

0. **Backlash $b_\gamma$ and time offset $t_0$:** As described in subsection 2.1, the dynamic parameters are assumed constant across the sky and can be estimated from a single Sun scan. We recommend averaging the results from multiple scans.

1. **Azimuth offset $\gamma_0$ and antenna tilt $\epsilon$:** Near the horizon, azimuth mispointing $\Delta\gamma$ can only be caused by $\gamma_0$ and $\epsilon$. The contributions differ between forward and reverse scans: $\gamma_0$ produces the same sign in both, while $\epsilon$ produces opposite signs in forward and reverse.

2. **East-West tilt $\alpha$ and elevation offset $\omega_0$:** Near the horizon in the East or West, elevation mispointing $\Delta\omega$ can only be caused by East-West pedestal tilt $\alpha$ or elevation offset $\omega_0$. In analogy to the previous step, forward and reverse scans allow these effects to be distinguished.

3. **North-South tilt $\delta$ and gimbal tilt $\beta$:** At high elevation in the South (or North in the Southern Hemisphere), elevation mispointing $\Delta\omega$ can be caused by North-South pedestal tilt $\delta$ or by $\omega_0$, which is already known from step 2. $\beta$ causes mainly an azimuth offset $\Delta\gamma$, but can also have a contribution to $\Delta\omega$.

4. $\alpha, \delta, \beta, \epsilon, \gamma_0, \omega_0, \chi$**:** Using the previous results as initial guesses, all parameters are jointly re-optimized against the full set of reference positions. At this stage, the flexibility parameter $\chi$ is included to account for residual elevation mispointings. Unlike $\alpha$, $\delta$, or $\omega_0$, the effect of $\chi$ depends on the absolute elevation.

Given a list of reference positions, the `SunscanPy` Python implementation automatically identifies the relevant sky regions and performs the sequential fitting procedure.

## 2.4 Step 3: Inverse Kinematics

Once an estimate of the pan-tilt system parameters $\mathcal{F}$ is available, misalignments can either be corrected mechanically—for example, by leveling the scanner pedestal—or compensated for in software. The latter requires inverting the forward model $M_\mathcal{F}$, such that for a desired celestial position $(\phi, \theta)$, the corresponding scanner axis positions $(\gamma, \omega)$ are determined:

$$(\gamma, \omega) = M_\mathcal{F}^{-1}(\phi, \theta). \tag{20}$$





This inversion can be performed numerically by minimizing the mispointing between the desired celestial coordinates and the beam position predicted by the forward model, i.e., by finding the scanner position $(\gamma, \omega)$ that minimizes

$$\Delta\Omega\big(M_{\mathcal{F}}(\gamma, \omega), (\phi, \theta)\big). \tag{21}$$

The feasibility of such inversion depends on the specific set of parameters. For example, offsets in the azimuth or elevation encoders $(\gamma_0, \omega_0)$ can be compensated straightforwardly. By contrast, an antenna tilt $\epsilon$ renders the zenith position unreachable, as will be discussed further in subsection 3.3

To test whether the inversion is successful, the obtained scanner position $(\gamma, \omega)$ is inserted back into the forward model, and the resulting pointing is evaluated against the target celestial position:

$$\Delta\Omega\big(M_{\mathcal{F}}(\gamma, \omega), (\phi, \theta)\big) \,=\, \Delta\Omega\big(M_{\mathcal{F}}(M_{\mathcal{F}}^{-1}(\phi, \theta)), (\phi, \theta)\big) \overset{!}{=} 0°. \tag{22}$$

## 3 Results

### 3.1 Step 1: Local Mispointing

Sun scans were carried out in August 2025 in Munich, southern Germany, using a Mira-35 Doppler cloud radar manufactured by Metek GmbH[2] (Görsdorf et al., 2015). The system operates in the Ka-band at $35.2$ GHz and is equipped with a $1.2$ m Cassegrain antenna. The radar is mounted on a steel platform at the top of the Munich Institute for Meteorology, supported by four adjustable anchor points. This setup provides a nearly unobstructed field of view down to $2°$ above the horizon. The antenna is mounted on a two-axis scanner with a nominal pointing precision of less than $0.1°$ in both azimuth and elevation. The signal transmission between the radar and control unit is performed via slip rings and a fiber optical rotary joint, which allows unrestricted continuous rotation in azimuth. The scanner joints operate in a temperature- and humidity-controlled environment, ensuring stable mechanical performance during the measurements.

In principle, Sun scans can be performed even in cloudy conditions and with the radar still emitting radiation. In this case, the sun signal is visible as enhanced receiver noise in target free sections along the beam, preferably in the far range. We focus on the radar range gates between $3$ km and $27$ km. We calculate the sun signal as the average noise level in all cloud free range gates. In general, for very low elevation angles, it is possible that there are not enough cloud free far range gates, in which case the scan is excluded from analysis. We performed all scans on almost cloud free days.

Figure 6a shows the measurements from a Sun scan performed on 19 August 2025 at 13:44 local time. The scan pattern follows the procedure described in Appendix A, with a radar averaging time of $0.3$ s. A small set of samples in the lower left corner were acquired to determine the background sky noise. The Sun is visible as an elongated region of enhanced signal strength. The two azimuth velocities used during the measurement, $0.2°\mathrm{s}^{-1}$ and $0.4°\mathrm{s}^{-1}$, are clearly visible as more separated data points for faster azimuth velocity. Depending on the direction of azimuthal motion, a "zigzag" shift of the solar signal becomes visible, which indicates either mechanical backlash or a timing offset in the scanner system.

---

[2]The instrument is a central component of the Munich ACTRIS National Facility for cloud remote sensing (https://nflabelling.actris.eu/facility/51); data and derived products are available via the Cloudnet portal https://cloudnet.fmi.fi/



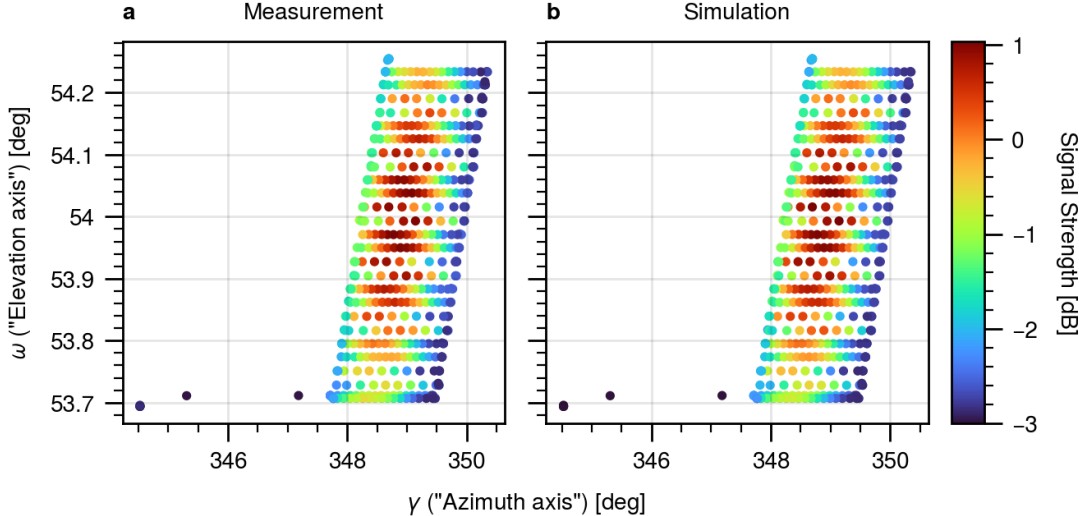

**Figure 6. a)**: Measurements of the Mira35 cloud radar when scanning the Sun. In azimuth, the scanner is alternating between two different azimuth velocities. **b)**: Simulated signal strength, using the parameters of Table 2. The parameters are determined as a best fit of the simulation to the measurements in a).

In Figure 6b, the optimal fit of the simulated signal strength to the measured signal is shown. We see that the simulated samples reproduce both the overall shape of the solar response and the azimuth-dependent displacement. The corresponding best-fit parameters are listed in Table 2.

With these parameters, the Sun's position can be transformed into beam-centered coordinates. Figure 7a and Figure 7b show the measured and simulated signal strength in this coordinate frame. Since the transformation incorporates the corrections for $\Delta\gamma$, $\Delta\omega$, $t_0$, and $b_\gamma$, the solar pattern appears undistorted and is centered at the origin. Figure 7c presents the residuals between measured and simulated signal strengths, with a mean deviation of $0.0013\,\mathrm{dB}(\pm 0.1065\,\mathrm{dB}$ standard deviation).

Repeating such Sun scans throughout the day allows us to assess the variability of the derived parameters. Figure 8 summarizes these results as a function of solar elevation angle. Figure 8a and Figure 8b show the beamwidths at half maximum in co- and cross-elevation directions. The cross-beamwidth is derived to approximately $0.55°$ across all scans, with a light $0.01°$ decreasing tendency for higher solar elevation angles. The co-beamwidth is on average about $0.515°$. There is an overall uncertainty of $\pm 1.5\,\%$ in both beamwidths.

Figure 8c and Figure 8d display the mispointing corrections $\Delta\gamma$ and $\Delta\omega$. Both exhibit systematic patterns over the course of the day, which will be further analyzed in subsection 3.2. Figure 8e shows the time offset, with $t_0 = -0.33$ s, indicating that the recorded signal corresponds to the scanner position approximately one-third of a second earlier. Such a timeshift is expected, since the Mira35 radar stores the time at the end of the $0.3\,\mathrm{s}$ averaging interval, plus an additional processing time of the central processing unit (CPU). A positive correlation between Sun elevation and time offset is observed, amounting to





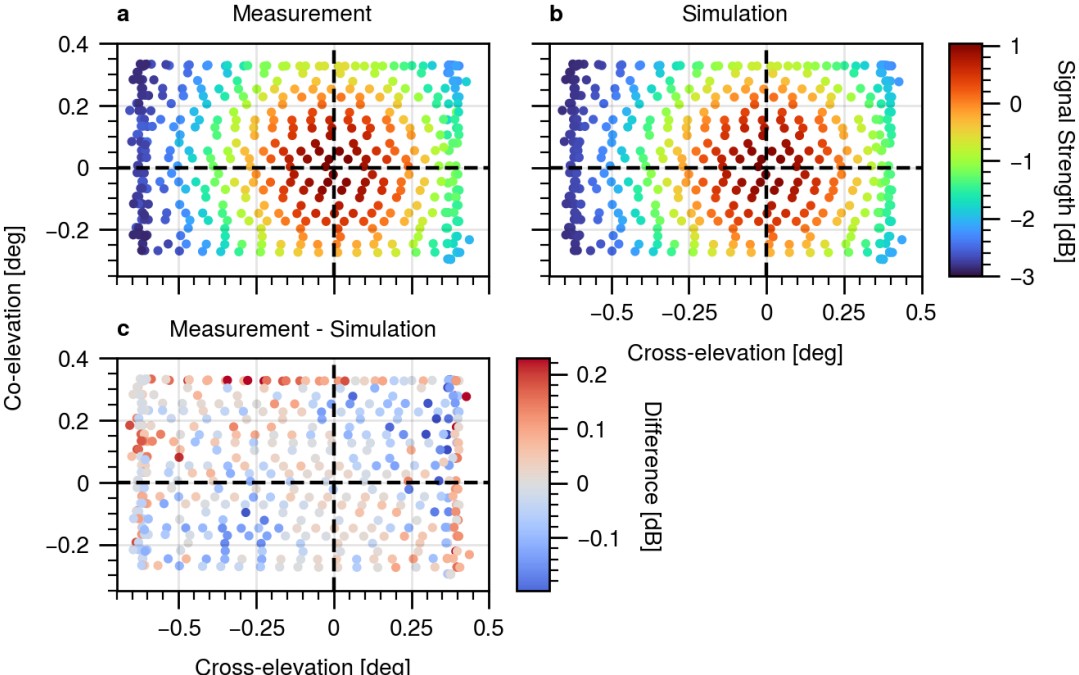

**Figure 7. a)** and **b)**: Same as Figure 6, but depicted in beam centered coordinates. **c)**: The difference between the signal strengths in a) and b).

a small change of approximately $0.03$ s over the full day. We suspect that this might be due to solar heating and resulting changes in temperature of the system, affecting the CPU processing times of signal and axis encoders. Importantly, the time offset accounts for more than $95\%$ of the dynamic azimuthal mispointing in the scans. The contribution from gear backlash

(Figure 8f) is negligible at $0.002°$.

The solar signal amplitude (Figure 8g) increases strongly from $0.8$ linear units ($-1.0$ dB) near the horizon to $1.5$ linear units ($1.8$ dB) at $20°$ solar elevation, reflecting the reduced atmospheric path length and hence lower path integrated gas attenuation of the direct solar radiation. At higher elevations, the changes in path length and signal are less significant. At the same time, the signal of the sky sample decreases slightly from $0.45$ linear units ($-3.5$ dB) by about $1\%$ to $0.445$ linear units from horizon

towards high noon.

### 3.2 Step 2: Scanner Inaccuracies

As described in subsection 2.2, each Sun scan provides a referenced pair of scanner and celestial coordinates $(\gamma_r, \omega_r)$ and $(\phi_r, \theta_r)$. With Sun scans distributed over the course of a full day, these data can be used to estimate the scanner inaccuracies following the procedure outlined in subsection 2.3.





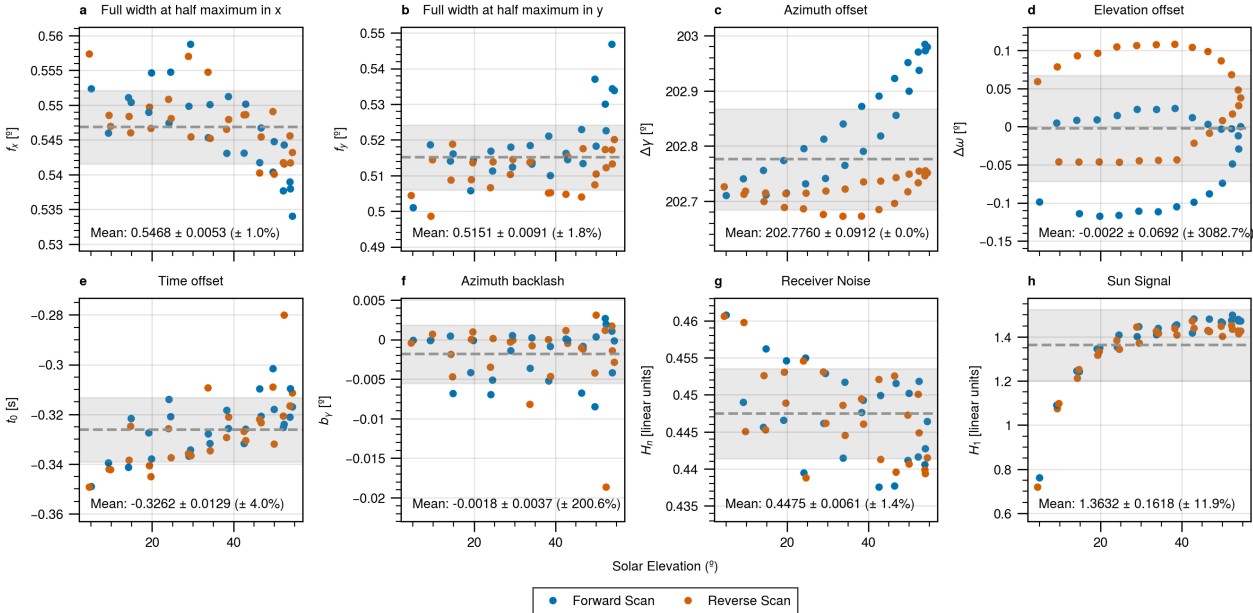

**Figure 8.** Variability of parameters over the course of one day, visualized as a function of solar elevation angle. The mean and standard deviation are indicated at the bottom of each panel and visualized as a dashed line and grey region, respectively.

| Parameter | Value |
|---|---|
| $\Delta\gamma$ [°] | 202.9727 |
| $\Delta\omega$ [°] | -0.0293 |
| $t_0$ [°] | -0.3097 |
| $f_x$ [°] | 0.5380 |
| $f_y$ [°] | 0.5343 |
| $b_\gamma$ [°] | -0.0042 |
| $H_n$ [dB] | -3.54 |
| $H_1$ [dB] | 1.68 |

**Table 2.** Parameters obtained from an optimal fit to the single Sun scan depicted in Figure 6 and Figure 7. Azimuth mispointing $\Delta\gamma$, elevation mispointing $\Delta\omega$, time offset between signal and scanner axis recording $t_0$, beam full width at half maximum in cross elevation $f_x$, beam full width at half maximum in co elevation $f_y$, scanner backlash $b_\gamma$, sky sample (receiver noise) $H_n$, sun brightness $H_1$.

Figure 9a presents Sun scans performed on 11 and 12 August 2025. The figure shows both the actual beam positions $(\phi_b, \theta_b)$ in the celestial hemisphere and the expected beam positions obtained from the scanner coordinates mapped into the sky using the identity scanner model,

$$(\phi_I, \theta_I) = M_\mathcal{I}(\gamma_r, \omega_r). \tag{23}$$





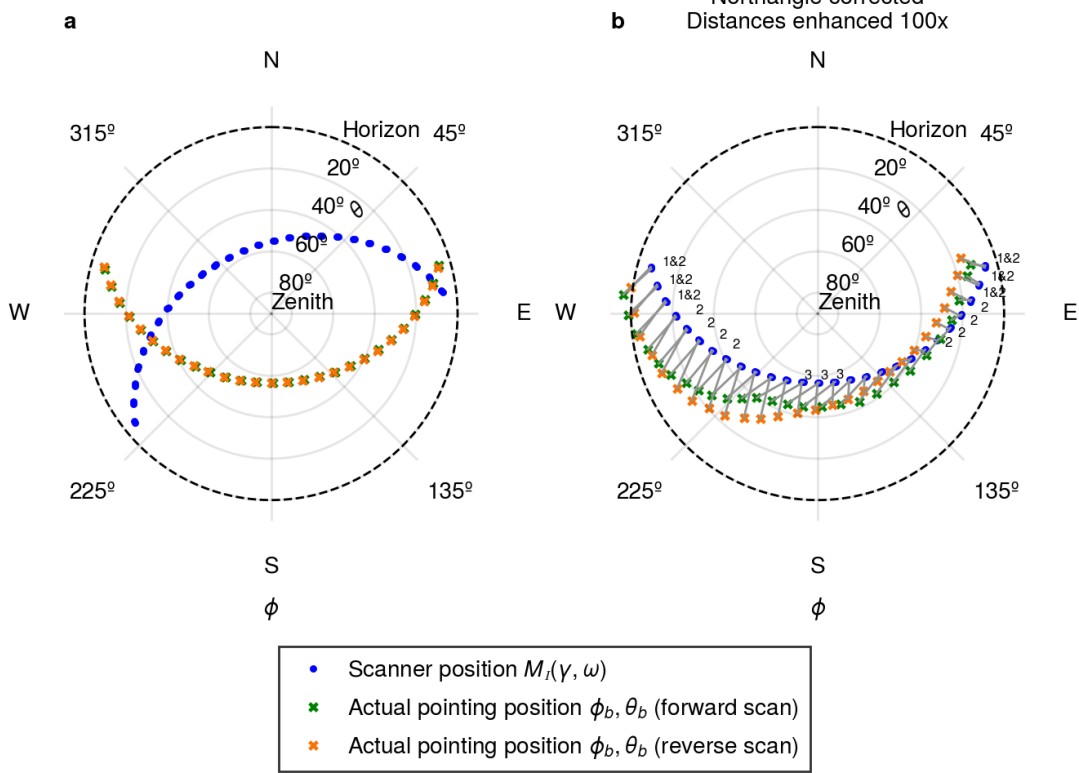

**Figure 9. a)**: Uncorrected scanner and beam positions, derived from multiple Sun scans over the course of one day. Note that for the reverse scans, the scanner elevation position is larger than $90°$. Therefore, to obtain spherical coordinates, the positions have to be transformed via the forward function $M_\mathcal{I}$ to a range $\phi \in [0°, 360°]$, $\theta \in [0°, 90°]$. **b)**: Same as a), but with a northangle correction applied to the scanner positions. The remaining differences between scanner and beam positions are very small. For better visualization, the differences are enhanced by a factor 100 using spherical linear extrapolation, and the beam position indicators are shifted accordingly along the grey lines.

The comparison reveals large discrepancies between expected and actual beam positions, due to an uncorrected north angle $\gamma_0 = 202.7°$. After adding the northangle, the residual differences are much smaller ($< 1°$), as shown in Figure 9b. For visibility, in Figure 9b, the residuals are exaggerated by a factor of 100 by spherical extrapolation of the actual beam positions.

Over the course of the day, the discrepancies exhibit a characteristic pattern. The largest deviations occur in the southwest, where the beam consistently points lower than expected. Since this effect is present in both forward and reverse scans, it indicates a pedestal tilt of approximately $0.2°$ towards the southwest.

We apply the sequential fitting procedure introduced in subsection 2.3 to the data in Figure 9. The numbers next to the points in Figure 9b indicate which samples are used for which step of the sequential fit. Table 3 lists the optimal scanner parameters





$\mathcal{F}_1$ obtained from this procedure. As expected, a positive $\alpha$ and negative $\delta$ confirm the southwest tilt. Compared to the pedestal tilt, the other inaccuracy parameters are about an order of magnitude smaller.[3]

| Parameters | Optimal Fit $\mathcal{F}_1$ | Optimal Fit $\mathcal{F}_2$ after manual leveling |
|---|---|---|
| $\gamma_0$ [°] | 202.7281 | 202.7491 |
| $\omega_0$ [°] | -0.0035 | -0.0087 |
| $\alpha$ [°] | 0.1123 | -0.0103 |
| $\delta$ [°] | -0.1259 | 0.0015 |
| $\beta$ [°] | -0.0927 | -0.0889 |
| $\epsilon$ [°] | 0.0110 | 0.0084 |
| $t_0$ [s] | -0.3247 | -0.3243 |
| $b_\gamma$ [°] | -0.0021 | -0.0031 |
| $\chi$ [°] | -0.0352 | -0.0494 |

**Table 3.** Scanner inaccuracies before and after manual leveling, as obtained from an optimal fit to multiple Sun scans. Azimuth offset $\gamma_0$, elevation offset $\omega_0$, West-East pedestal tilt $\alpha$, North-South pedestal tilt $\delta$, gimbal tilt $\beta$, antenna tilt $\epsilon$, signal-scanner time offset $t_0$, scanner backlash $b_\gamma$, elastic elevation deformation $\chi$.

Figure 10 shows the expected scanner positions when the optimal parameters $\mathcal{F}_1$ are taken into account. Relative to Figure 9b,
the discrepancies are reduced by a factor of seven, yielding an average residual of only $0.02°$. The remaining deviations are likely caused by effects not explicitly represented in the scanner model, such as position-dependent offsets, nonlinear elastic deformations, or differential thermal expansion of the steel support structure during the day.

With precise knowledge of the pedestal misalignment, we can proceed to correct it mechanically. Our radar is mounted on four adjustable screws arranged in a $977\,\mathrm{mm}$ by $1328\,\mathrm{mm}$ rectangle. Each full turn of a screw raises the corresponding corner
by $2\,\mathrm{mm}$, enabling fine adjustments. Figure 11a presents the results from a second series of Sun scans after adjusting the feet to compensate for the pedestal tilt. The remaining discrepancies are now dominated by other scanner inaccuracies, such as motor offsets, gimbal tilt $\beta$, and antenna tilt $\epsilon$. As a consistency check, we re-estimated the optimal scanner parameters $\mathcal{F}_2$ from these new scans. The results, shown in Table 3 (second column), confirm that the pedestal is now aligned to within $0.01°$, while the other parameters remain similar to their previous values.

---

[3]These comparably small axis offsets, gimbal tilt and antenna tilt are to be expected, since the scanner had undergone a factory calibration prior to the measurements shown in Figure 9. The factory procedure is based on a manual Sun scan analysis. For the experiment presented here, the pedestal was intentionally misaligned to test the fitting method.





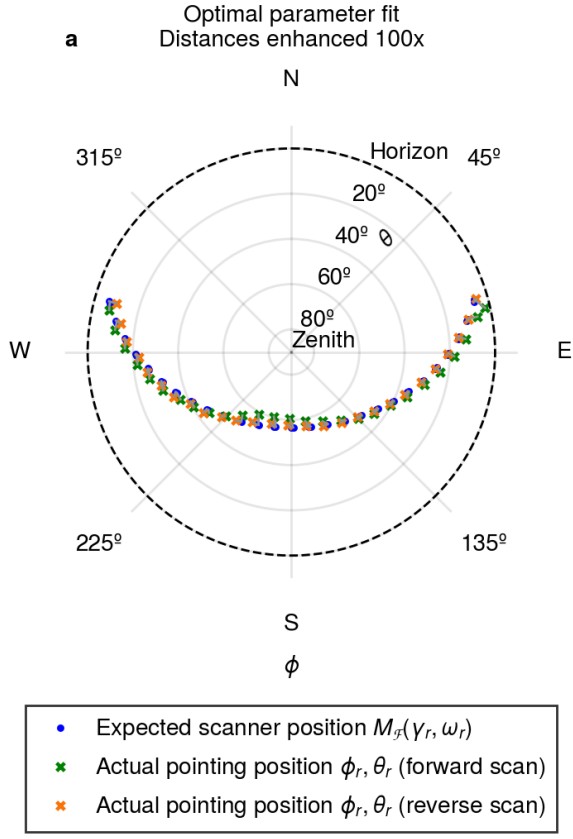

**Figure 10.** Same as Figure 9, but with the expected beam position calculated from the scanner axes coordinates based on the fitted model $M_{\mathcal{F}1}$, instead of just transforming to spherical coordinates via $M_{\mathcal{I}}$.

## 3.3 Step 3: Inverse Kinematics

Instead of correcting the pointing errors mechanically, the ability to invert the scanner model $M_{\mathcal{P}}$ allows for an automatic correction in software. For example, if the radar is required to point to $(\phi = 0°, \theta = 30°)$ in the sky, the corresponding scanner coordinates according to the fitted parameter set $\mathcal{F}_1$ are:

$$M_{\mathcal{F}1}^{-1}(\phi = 0°, \theta = 30°) = \begin{cases} \gamma = 157.30°, & \omega = 29.91°, \\ \gamma = 337.38°, & \omega = 150.10°. \end{cases} \tag{24}$$

The two solutions correspond to the forward and backward scanner configurations. Both solutions account for the complete set of inaccuracies represented in the scanner model. For cloud radar applications, the zenith position is typically the most critical. In this case, the inversion yields:



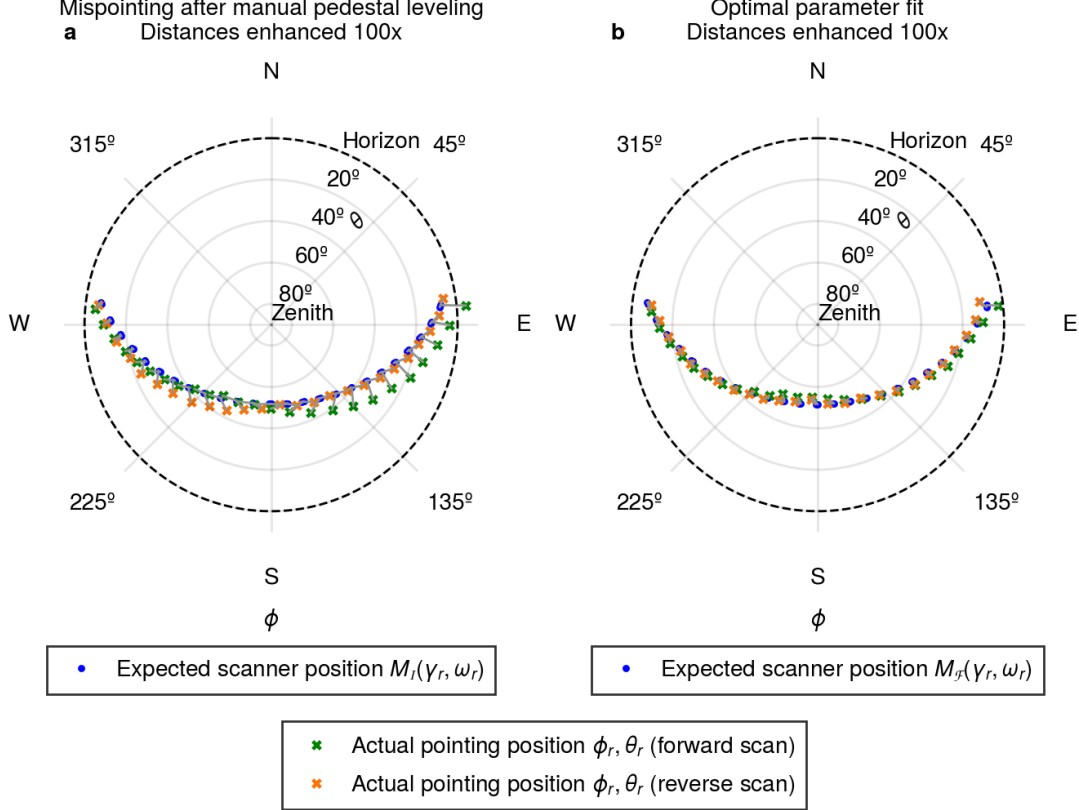

**Figure 11. a)**: Scanner position and actual beam pointing after manually leveling the pedestal. **b)**: Same as a), but instead of just transforming the scanner coordinates to spherical coordinates via $M_{\mathcal{I}}$, use the fitted model $M_{\mathcal{F}2}$.

$$M_{\mathcal{F}1}^{-1}(\phi = 0°, \theta = 90°) = \begin{cases} \gamma = 171.06°, & \omega = 89.85°, \\ \gamma = 47.47°, & \omega = 90.15°. \end{cases} \tag{25}$$

At zenith, the prescribed azimuth $\phi = 0°$ is irrelevant for the pointing accuracy. The scanner azimuth positions $\gamma$ listed in
Equation 25 were chosen to optimize the final pointing.

Figure 12a and Figure 12b illustrate the corrections required to be added to the scanner coordinates $\gamma$ and $\omega$, respectively, to achieve perfect pointing across the full celestial hemisphere. Figure 12a is dominated by the system's north angle of $157.3°$. Near zenith, the azimuth corrections increase, reflecting the reduced effectiveness of azimuth corrections at high elevation angles. Figure 12b is dominated by the southwest tilt of the pedestal: in the southwest, the elevation axis must be increased by
about $0.2°$, while in the northeast it must be reduced by roughly $0.1°$.



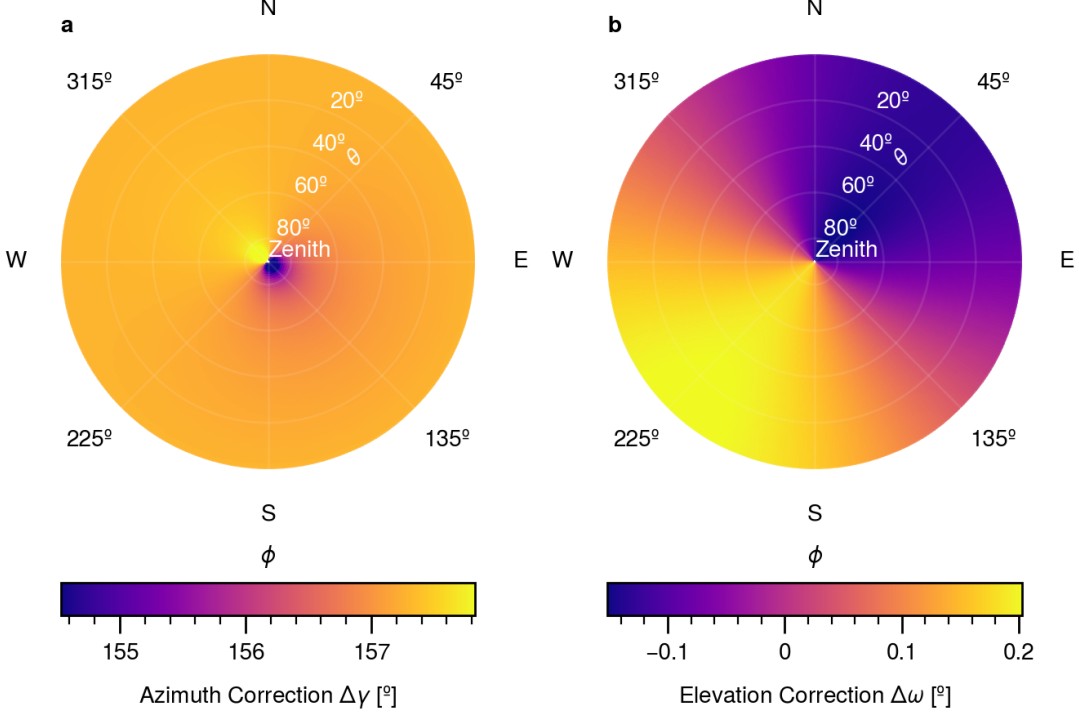

**Figure 12.** Corrections to be added to the scanner azimuth and elevation axis position in order to achieve perfect beam pointing accuracy, for all locations of the celestial hemisphere. **a**): Azimuth correction. **b**): Elevation correction.

It should be noted that inverse kinematics does not always allow for complete correction. Certain scanner configurations restrict accessibility to specific sky regions. For illustration, consider the extreme case where the dish is tilted by $\epsilon = 90°$, effectively aligning the beam with the elevation axis. In this configuration, the elevation motor no longer affects the beam pointing, making it obviously impossible to reach the zenith position. Figure 13a shows the hypothetical case of an antenna tilt of $\epsilon = 10°$. Here, the maximum achievable beam elevation is $80°$, leaving the zenith position unreachable. Above this limit, the residual mispointing after inverse kinematics correction increases continuously, reaching up to $10°$ at zenith.

A straightforward solution for Doppler cloud radars is to deliberately shift the unreachable polar region away from zenith by tilting the pedestal. Figure 13b illustrates this effect for a virtual scanner tilted by $6°$ towards both the west and north. The unreachable region is displaced accordingly, and zenith can again be reached with nearly perfect accuracy. In fact, the same result would be achieved for a sufficiently large pedestal tilt into any direction. This leads to the paradoxical effect that with the ability to perform inverse kinematics corrections, a perfectly leveled pedestal represents the least favorable configuration, as it maximizes the risk of an unreachable zenith.





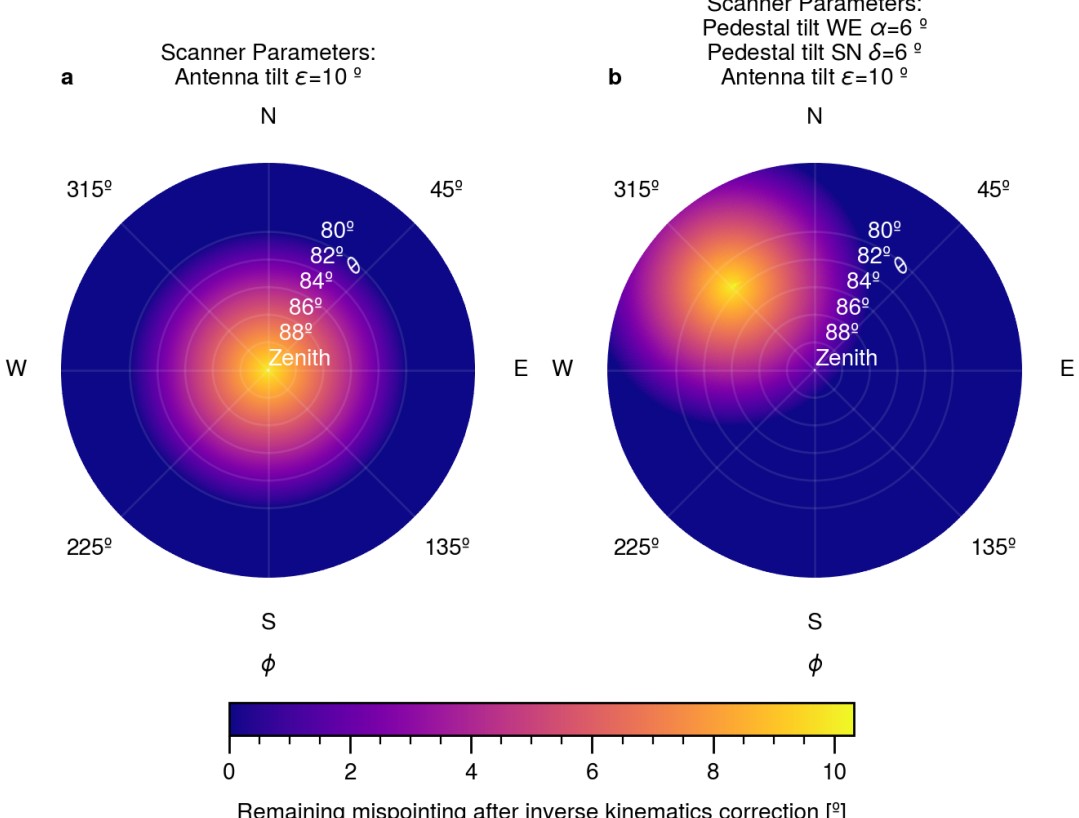

**Figure 13.** Simulated pointing deviation from desired location after inverse kinematics correction for two different scanner configurations. **a)**: A perfectly leveled scanner with an antenna tilt of $10°$. **b)**: Like a), but with the scanner pedestal tilted towards North and West by $6°$

## 3.4 Pointing Uncertainty

Figure 8 already provides a measure of the relative uncertainty of the Sun scan parameters by showing the variability of repeated
scans throughout the day. Beyond this variability, sources of absolute bias must also be considered.

A potential source for uncertainty is the accuracy of the solar position algorithm. We compared four different algorithms. `Skyfield` (Rhodes, 2019), `Sunpy` (The SunPy Community et al., 2020) and `PySolar` (Stafford, 2021) are free and open source Python modules. `SolCalc` (NOAA Global Monitoring Laboratory, 2025) is the solar position calculator from the US National Oceanic and Atmospheric Administration. We found general agreement in the solar azimuth and elevation of better
than $0.02°$ between the libraries. The main cause for this difference is whether refraction is considered and how a correction is implemented. For the calculations in this manuscript and `SunscanPy`, we rely on the geometrical (i.e. not refraction corrected) position provided by `Skyfield`. We then apply the microwave refraction formula proposed by Huuskonen and



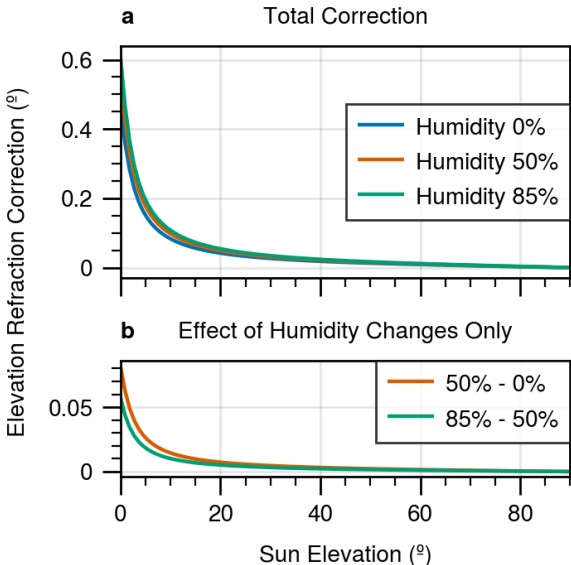

**Figure 14.** Apparent shift in elevation of the solar disk due to atmospheric refraction of GHz radio waves, as a function of solar elevation. **a**): Absolute shift. **b**): Differences between the curves in a), with the $0\%$ humidity curve taken as a reference.

Holleman (2007). Their coefficients were fitted to simulations of the Starlink positional astronomy library (Currie et al., 2014), which uses the refractivity model from Rueger (2002a). The Starlink implementation is generally valid for microwave radiation

in the GHz range, with a reported accuracy better than $1''$. It does not include an explicit wavelength dependency. According to Rueger (2002b), dispersion of the refractive index would be most relevant near oxygen and water vapor resonance lines, which are deliberately outside of the frequency bands used by weather and cloud radars. Figure 14a shows the correction to the geometric solar elevation as a function of Sun elevation. The effect is largest near the horizon, but already at elevations above $10°$, the correction remains below $0.1°$. The dominant source of uncertainty in this parameterization is atmospheric

humidity. Figure 14b illustrates how the apparent solar elevation changes when relative humidity increases from $0\%$ to $50\%$, and from $50\%$ to $85\%$. Above $10°$, the associated effect is less than $0.01°$. For the calculations in the previous sections, we assumed a constant humidity of $50\%$. While future refinements could incorporate vertical humidity profiles from radiosonde measurements, this level of detail is not necessary to achieve an absolute calibration accuracy well below $0.1°$. This conclusion is confirmed by sensitivity tests: repeating our calculations with $85\%$ humidity altered the derived scanner parameters by in

general less than $10''$.

In order to estimate the sun position accurately, precise timing is important. As described in subsection 2.1, we use different azimuth velocities to estimate the relative time offset between signal and scanner encoder readings. However, this method can not correct for an absolute time offset being present in signal processor and scanner encoder alike. Absolute time offsets will cause an apparent mispointing, which is always directed along the trajectory of the sun. Depending on the latitude and season,





it may therefore be possible to derive an absolute time offset from the characteristic misalignment pattern of multiple Sun scans. Nevertheless, for practical applications, we recommend to simply synchronize the time of the radar before a Sun scan calibration. Using for example GPS time or the standard network time protocol (NTP) synchronization available in modern operating systems, an accuracy of $100\,\mathrm{ms}$ or better should be easily achieved (Mills, 1994). From sensitivity tests, we found time shifts of $1\,\mathrm{s}$ to cause in general less than $10\,''$ shift in the derived scanner parameters.

## 4 SunscanPy Python Implementation

The calibration framework developed in this study is provided as an open-source Python library SunscanPy. The library is written in a modular, object-oriented style and reads data in the form of one dimensional numpy arrays. This facilitates the inclusion of the methods into existing data piplines and operational environments. Figure 15 provides a schematic overview of the data processing. This processing chain was also used for the results in this study. Together with the library, we pro-
vide two detailed tutorial notebooks covering signal estimation and scanner estimation, guiding users through the full solar pointing calibration process. In addition, the library includes various utility functions for atmospheric refraction correction, solar position calculation, Sun scan data plotting, and an interactive 3D visualization of the scanner geometry. For usage, the recommended starting point is the project homepage at https://github.com/Ockenfuss/sunscanpy, where the tutorial notebooks are also provided.

## 5 Conclusions

We have presented a generic theory and workflow for radar pointing calibration using the Sun as a reference source. The procedure is structured into three major steps: (i) recording and evaluation of individual Sun scans, (ii) derivation of scanner inaccuracies, and (iii) correction of these inaccuracies. In total, the method allows the derivation of 13 distinct parameters. These include the radar beamwidth in co- and cross-elevation directions, the scanner pedestal tilt, axis misalignments, encoder
offsets, gear backlash, and the receiver-scanner time offset. Our results demonstrate that, with this approach, a pointing accuracy on the order of $0.01°$ can be achieved.

Although the workflow is demonstrated here with a Mira-35 cloud radar, the methodology has been deliberately designed to be radar-agnostic. The procedures apply to any instrument equipped with a parabolic antenna mounted on a two-axis scanner, whether the radar is a cloud research radar or used for weather surveillance.
To facilitate the application, we provide a novel open-source Python package SunscanPy, which contains a suite of tools together with comprehensive tutorials and example data for radar pointing calibration. We also discuss the possibility of fully automatic radar pointing calibration. SunscanPy implements an inversion method for the scanner model, enabling calculation of the scanner coordinates required to achieve a desired celestial pointing. This opens the possibility for active radar pointing calibration as an alternative to manual, mechanical adjustments: after performing Sun scans, a radar can infer its scanner mis-





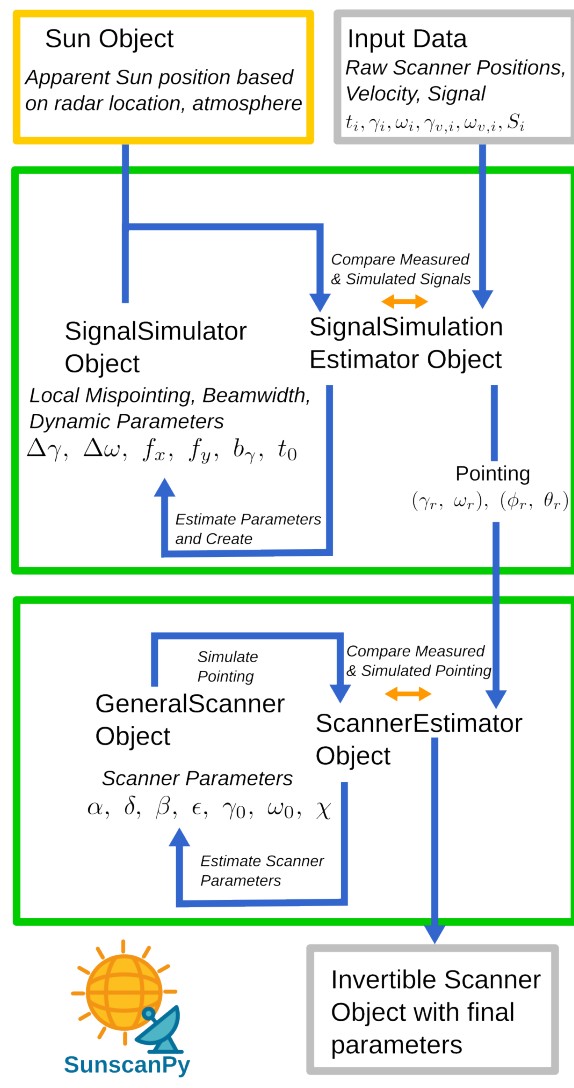

**Figure 15.** Overview of the modules and dataflow in the `SunscanPy` Python library. The upper and lower green boxes correspond conceptually to step one and two in Figure 1, respectively.

alignment and then compensate for any bias in subsequent measurements. Such functionality will be highly valuable for mobile radars, for example deployed in field campaigns, where uneven ground or snow may cause significant pedestal misalignment.

Stationary scanning radars likewise benefit from our workflow. `SunscanPy` makes it possible to monitor pointing accuracy over long time scales, and, given knowledge of the scanner configuration, to retrospectively improve the pointing accuracy of existing datasets. For these reasons, we think that our tool will be of interest to network-level initiatives such as the Euro-

pean Aerosol, Clouds and Trace Gases Research Infrastructure (ACTRIS, Laj et al. (2024)) or the U.S. Atmospheric Radiation





Measurement (ARM, Mather and Voyles (2013)) program. Long-term cloud radar datasets are increasingly used to infer statistics of, for example, hydrometeor sedimentation velocities or vertical air motion (e.g., Kalesse and Kollias, 2013). Also new satellite missions, such as EarthCare, benefit from well-calibrated ground-based radar datasets to evaluate their novel Doppler velocity capabilities.

Last, but not least, while not investigated in this study, we believe that our methodology and the `SunscanPy` package can also be applied to other scanning, two-axis instruments like sun photometers or microwave radiometers. Particularly step (ii) and (iii) of the workflow have the potential to be transferred with minor to no adoptions to those instruments.

*Code and data availability.* The code for the full Sun calibration procedure is made freely available in the form of an open source Python package `SunscanPy` at https://github.com/Ockenfuss/sunscanpy. The package includes some example datasets as used in this publication,
as well as plotting functionality. There are two extensive tutorial jupyter notebooks to guide through the calibration and visualization process at the package homepage.

## Appendix A: Implementation of a Sun Scan Pattern

As explained in subsubsection 2.2.2, any scanning pattern which yields sufficient samples around the sun can be evaluated. For practical applications, some patterns are more efficient in terms of scanning time and spatial resolution. Given the huge
variety of hardware and programming languages used to control scanning weather and cloud radars, it is difficult to provide ready-to-run scripts for every radar. Instead, in algorithm A1 we provide a pseudocode implementation of the scan pattern we used for our measurements. The only radar specific command in the pseudocode is MOVE TO, which steers the scanner to a given pair of axis positions. If such a command is available, it should be straightforward to implement a Sun scan in any programming language. Figure A1 illustrates the resulting pattern. Our code takes the movement of the Sun into account and
dynamically follows the expected position of the solar disk. If the Sun is located at high elevations in the sky, the azimuth range and azimuth speed are enhanced to sample a constant solid angle in the sky. To perform a reverse scan, the only adaption is to exchange the MOVE TO function in algorithm A1 with the MOVE REVERSE function described in algorithm A2.



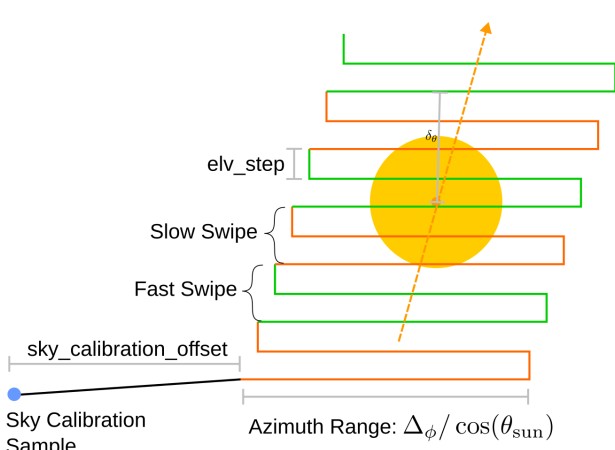

**Figure A1.** Illustration of the scan pattern used in Figure 6. Algorithm A1 gives an example how this can be implemented in code.




---

**Algorithm A1** Sun scan pattern algorithm

---

**Require:** $\Delta_\phi$, $\Delta_\theta$, elv_step, speed_fast, speed_slow, sky_calibration_offset

  **Input parameters:**

    $\Delta_\phi$: Azimuth width of the search pattern

    $\Delta_\theta$: Elevation width of the search pattern

    elv_step: Step width in elevation

    speed_fast: Speed in azimuth for the fast swipes

    speed_slow: Speed in azimuth for the slow swipes

    sky_calibration_offset: Additional offset in azimuth to measure the sky calibration sample

 

  $\phi_{\text{sun}}, \theta_{\text{sun}} \leftarrow$ compute_sun_location()

  azimuth_factor $\leftarrow 1/\cos(\theta_{\text{sun}})$        *// correct the azimuth range for the elevation of the sun*

  azimuth_factor $\leftarrow \min(\text{azimuth\_factor}, 4)$        *// limit to avoid too large speeds and ranges if the sun is close to zenith*

  $\Delta_\phi \leftarrow \Delta_\phi \times$ azimuth_factor

  speed_fast $\leftarrow$ speed_fast $\times$ azimuth_factor

  speed_slow $\leftarrow$ speed_slow $\times$ azimuth_factor

  sky_calibration_offset $\leftarrow$ sky_calibration_offset $\times$ azimuth_factor

 

  **move to** $(\phi_{\text{sun}} - \Delta_\phi - \text{sky\_calibration\_offset}, \theta_{\text{sun}} - \Delta_\theta)$    *// move to the sky calibration position*

  **sleep**(1)    *// wait one second to measure some sky background values*

  $\phi_{\text{sun}}, \theta_{\text{sun}} \leftarrow$ compute_sun_location()    *// update sun position*

  $\delta_\theta \leftarrow -\Delta_\theta$    *// current elevation of the beam relative to the (moving) sun*

  **move to** $(\phi_{\text{sun}} - \Delta_\phi, \theta_{\text{sun}} + \delta_\theta)$    *// move to the exact starting position*

  (Algorithm continues on next page)

---





---

**Algorithm A1** Sun scan pattern algorithm (continued)

---

iteration $\leftarrow 0$

**while** $\delta_\theta < \Delta_\theta$ **do**

    **if** iteration mod $2 = 0$ **then**

        $\phi_v \leftarrow$ speed_slow

    **else**

        $\phi_v \leftarrow$ speed_fast

    **end if**

    **move to** $(\phi_\text{sun} + \Delta_\phi, \theta_\text{sun} + \delta_\theta)$ **at speed** $\phi_v$         *// move right*

    $\phi_\text{sun}, \theta_\text{sun} \leftarrow$ compute_sun_location()         *// update sun position*

    $\delta_\theta \leftarrow \delta_\theta +$ elv_step         *// update relative elevation*

    **move to** $(\phi_\text{sun} + \Delta_\phi, \theta_\text{sun} + \delta_\theta)$ **at speed** $\phi_v$         *// move up*

    **move to** $(\phi_\text{sun} - \Delta_\phi, \theta_\text{sun} + \delta_\theta)$ **at speed** $\phi_v$         *// move left*

    $\phi_\text{sun}, \theta_\text{sun} \leftarrow$ compute_sun_location()         *// update sun position*

    $\delta_\theta \leftarrow \delta_\theta +$ elv_step         *// update relative elevation*

    **move to** $(\phi_\text{sun} - \Delta_\phi, \theta_\text{sun} + \delta_\theta)$ **at speed** $\phi_v$         *// move up*

    iteration $\leftarrow$ iteration $+ 1$

**end while**

---

**Algorithm A2** Move reverse function

---

**Require:** $\gamma, \omega$

    **Input parameters:**

        $\gamma$: Desired forward azimuth position in the sky.

        $\omega$: Desired forward elevation position in the sky.

$\gamma_\text{reverse} \leftarrow (\gamma + 180°) \mod 360°$

$\omega_\text{reverse} \leftarrow 180° - \omega$

**move to** $(\gamma_\text{reverse}, \omega_\text{reverse})$

---

**List of Symbols**

$\alpha$   Pedestal tilt towards West or East

$\beta$   Gimbal tilt

$b_\gamma$   Azimuth backlash





$\delta$  Pedestal tilt towards North or South

$\epsilon$  Antenna tilt

$\gamma$  Azimuth axis position

$\gamma_0$  Azimuth axis offset

$\dot{\gamma}$  Azimuth axis velocity

$\omega$  Elevation axis position

$\omega_0$  Elevation axis offset

$\dot{\omega}$  Elevation axis velocity

$t_0$  Time offset between axis positioners and signal recorder

$\chi$  Elastic elevation deformation

$\theta$  Elevation angle (celestial coordinates)

$\phi$  Azimuth angle (celestial coordinates)

$\phi_s$  Solar azimuth position

$\theta_s$  Solar elevation position

$\phi_b$  Beam azimuth position

$\theta_b$  Beam elevation position

$\Delta\gamma$  Local azimuth mispointing

$\Delta\omega$  Local elevation mispointing

$\Delta\Omega$  Angular mispointing

$f_x$  Beam full width at half maximum in cross-elevation direction

$f_y$  Beam full width at half maximum in co-elevation direction

$G$  Antenna beam pattern function

$H$  Solar emission pattern function

$H_0$  Sky brightness



$H_1$  Solar disk brightness

$H_n$  Receiver noise level

$Q$  Simulated signal strength

$S$  Measured signal strength

$r_s$  Angular radius of solar disk

$M_{\mathcal{P}}$  Forward scanner model with parameters $\mathcal{P}$

$M_{\mathcal{I}}$  Ideal scanner model

$\mathcal{P}$  Set of scanner inaccuracy parameters

$\mathcal{F}$  Fitted scanner parameters

$J_1$  First-order Bessel function

$b_x, b_y, b_z$  Beam-centered coordinate axes

$x_0, y_0$  Beam shape coefficients

$r_{0.5}$  Radius where beam has half maximum value

*Author contributions.* PO developed the method and `SunscanPy` , evaluated the measurements and wrote this publication. PO and GK
performed the measurements. GK also co-authored `SunscanPy` and this publication. GK is responsible for the general Mira35 radar
operation and data quality. MB and SK assisted with longterm Sun scan and radar pointing experience. Some concepts, namely the reverse
scan and the scanner inaccuracies, are based on earlier work from MB. All authors edited and proofread the manuscript.

*Competing interests.* MB is employed by METEK Meteorologische Messtechnik GmbH, the manufacturer of the Mira35 cloud radar.

*Acknowledgements.* This work has been supported by the DFG Priority Program SPP2115 "Fusion of Radar Polarimetry and Numerical At-
mospheric Modelling Towards an Improved Understanding of Cloud and Precipitation Processes" (PROM) under Grant PROM-POMODORI
(Project Number 408012686) and PROM-ICEPOLCKA (Project Number 408027579). We also acknowledge support for hardware upgrades
and long-term operation of the MIRA35 system by ACTRIS-D (Grant number 01LK2001E), funded by the Federal Ministry of Education
and Research (BMBF) under the FONA Strategy "Research for Sustainability." In the production of this study, tools based on artificial intel-
ligence (AI) were used. Specifically, "ChatGPT" by OpenAI and "Claude" by Anthropic assisted in simple, repetitive code generation tasks





(plotting and formatting), aw well as in improving the wording and formulation of this manuscript. All ideas and concepts are developed without AI assistance exclusively by the authors.



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
