# Peer review of "A Novel Framework for Automatic Scanning Radar Pointing Calibration Using the Sun"

_EGUsphere, 2025_

## Referee Comment (RC1)

**Review - A Novel Framework for Automatic Scanning Radar Pointing Calibration Using the Sun**

Paul Ockenfuß, Gregor Köcher, Matthias Bauer-Pfundstein, and Stefan Kneifel

The authors present a novel framework to correct antenna beam pointing of scanning cloud or weather radar by using the sun as a reference target. The paper describes the workflow, which is also made publicly available through the open-source SunscanPy python package. The presented approach is demonstrated using a 35 GHz cloud radar and the authors show that inverse kinematics can be used to automatically correct pointing. With their approach, an absolute pointing accuracy better than 0.1 degrees can be achieved.

Given the importance of pointing accuracy, SunscanPy offers a highly valuable contribution to the field. The paper is well written and the authors manage to explain the technical framework clearly. I don't have major concerns and would therefore recommend publishing this paper after the following minor comments are addressed.

**General comments**

- Although I commend the authors on readability given the technical nature of the manuscript, I think the visual presentation could be improved. Figures should be understandable "at first glance". At the moment, some figures need quite some time to fully take in. Readers should know what is shown with the figure caption alone and all parameters / abbreviations should be clear. I understand that due to the content, many parameters are necessary and it might be too detailed to list all in each figure. For efficiency, something along the lines of "for nomenclature, see text" could be added to the respective captions. Also, subpanels should be labeled consistently across the manuscript. Some text (e.g., parameters in Fig. 1) are very small. Please double check if everything is readable in a realistic size at 100%. Multiple figures could be improved in my opinion to make the paper more accessible / easier to follow. I find it hard to read the colorbars in Fig. 4 (see specific comments). I would argue that Fig. 5 is not clear enough and should include labels of angles / parameters or be more explicit what the arrows show. I wonder if Fig. 9-11 could be combined into one; or possibly 10-11. I also think that Fig. 15 could be adapted to increase readability. For example, it could be noted in the figure what the boxes represent and the text corresponding to blue arrows could be color-coded blue as well such that is clear what text belongs to where. Currently at first glace everything blends together and it takes quite long to follow the flow / logic of the figure. I think there's room to improve without too much work.

- It is not clear to me what the limitations of the approach in terms of latitude / seasonality are if any. Can you include recommendations and/or make it explicit whether this can be applied for example in polar regions? Given the dependence on sun elevation, I assume there might be ambiguities, but maybe this is no issue.

**Specific comments**

- Line 25: as is common in the higher atmosphere → include a reference
- Line 38: The benefits of solar calibration are now widely acknowledged → I would also expect some references here
- Line 75: applicable by → is that the right English term? I'm also not a native speaker
- Line 83: since SunscanPy is first mentioned here: I would recommend to publish the code on Zenodo or similar to get a DOI. It can then be directly referenced here
- Line 177: Out of curiosity, how accurate is it to model the solar emission pattern as a bipartite function? Maybe include some references
- Line 185: What about other radar systems?
- Figure 4:
  - insensitive → do you mean less sensitive?
  - grey-white colorbar → I get the idea, but I can't tell when it is blue / yellow which greyscale that corresponds to. can you please include blue and yellow color bars instead?
  - I understand the red line is the expected signal resulting from background, beampattern, and sun. However, I find it unintuitive how you depict this currently, since you show a discrete colorbar with 0.0 and 1.8 dB and draw 0.8 in the middle. I suggest to remove and only keep in the figure caption or find a more intuitive way to include in the plot.
- Line 254: Nelder-Mead simplex optimization → can you include a reference?
- Line 260: How many scans do you recommend to average?
- Line 305: small number of samples → please be more ecplicit. How many are a small number?
- Footnote page 15:I would also include a link to the specific data you used using the custom DOI feature of cloudnet (at least in the data availability section)
- Figure 9: the grey lines should also be included in the legend
- Line 430: I had a look at the tutorial notebooks (very nice!). Maybe it would help to also include a sketch with the different angle/parameter definitions in part 1

**Technical corrections**

- Line numbers: there are more than 5 lines between 90 and 95. Please correct to make the review process easier in case there is a second round after revisions.

- Equations: Please check the punctuation after equations (e.g., 1, 2, 9, 10, 13, 14). I would also give MP: (γ, ω) →(φ, θ) a number, or put in text.
- Line 3: particle's → particles'
- Line 75: applicable by → is that the right English term? I'm also not a native speaker
- Line 80 and onwards: Section instead of section
- Line 8:1 sun → be consistent if sun or Sun
- Figure 4: puse → pure
- Line 285: period at the end missing
- Figure 10: has an "a" label, although there are no other panels.
- Line 428: piplines → pipelines
- Line 455-456: fix citation style (remove brackets around year)
- Line 530: aw -> as